# Concurrent exercise training induces additional benefits to hydrochlorothiazide: Evidence for an improvement of autonomic control and oxidative stress in a model of hypertension and postmenopause

**Maycon Junior Ferreira[1], Michel Pablo dos Santos Ferreira Silva[2], Danielle da Silva Dias[3,4], Nathalia Bernardes[5], Maria Claudia Irigoyen[3], Kátia De Angelis[1,2]***

**1** Exercise Physiology Laboratory, Universidade Federal de São Paulo (UNIFESP), São Paulo, SP, Brazil,
**2** Translational Physiology Laboratory, Universidade Nove de Julho (UNINOVE), São Paulo, SP, Brazil,
**3** Hypertension Unit, Heart Institute (InCor), School of Medicine, Universidade de São Paulo (USP), São Paulo, SP, Brazil, **4** Postgraduate Program in Physical Education, Universidade Federal do Maranhão, São Luís, MA, Brazil, **5** Human Movement Laboratory, Universidade São Judas Tadeu (USJT), São Paulo, SP, Brazil

* prof.kangelis@yahoo.com.br

## Abstract

### Objective

This study aimed to evaluate whether exercise training could contribute to a better modulation of the neurohumoral mechanisms linked to the pathophysiology of arterial hypertension (AH) in postmenopausal hypertensive rats treated with hydrochlorothiazide (HCTZ).

### Methods

Female spontaneously hypertensive rats (SHR) (150−200g, 90 days old) were distributed into 5 hypertensive groups (n = 7–8 rats/group): control (C), ovariectomized (O), ovariectomized treated with HCTZ (OH), ovariectomized submitted to exercise training (OT) and ovariectomized submitted to exercise training and treated with HCTZ (OTH). Ovarian hormone deprivation was performed through bilateral ovariectomy. HCTZ (30mg/kg/day) and concurrent exercise training (3d/wk) were conducted lasted 8 weeks. Arterial pressure (AP) was directly recorded. Cardiac effort was evaluated using the rate-pressure product (RPP = systolic AP x heart rate). Vasopressin V1 receptor antagonist, losartan and hexamethonium were sequentially injected to evaluate the vasopressor systems. Inflammation and oxidative stress were evaluated in cardiac tissue.

### Results

In addition to the reduction in AP, trained groups improved RPP, AP variability, bradycardic (OT: −1.3 ± 0.4 and OTH: −1.6 ± 0.3 vs. O: −0.6 ± 0.3 bpm/mmHg) and tachycardic responses of baroreflex sensitivity (OT: −2.4 ± 0.8 and OTH: −2.4 ± 0.8 vs. O: −1.3 ± 0.5

**Data Availability Statement:** All relevant data are within the paper and its Supporting Information files.

**Funding:** MJF: 2019/06277-0, São Paulo Research Foundation (FAPESP) (https://fapesp.br/). KDA and MCI: National Council for Scientific and Technological Development (CNPq) (407398/ 2021-0; 406792/2022-4) (https://www.gov.br/ cnpq/pt-br). Kátia De Angelis and Maria Claudia Irigoyen are recipients of CNPq Fellowship (CNPq-BPQ). The funders had no role in study design, data collection and analysis, decision to publish, or preparation of the manuscript.

**Competing interests:** The authors have declared that no competing interests exist.

bpm/mmHg), NADPH oxidase and IL-10/TNF-α ratio. Hexamethonium injection revealed reduced sympathetic contribution on basal AP in OTH group (OTH: −49.8 ± 12.4 vs. O: −74.6 ± 18.1 mmHg). Furthermore, cardiac sympathovagal balance (LF/HF ratio), IL-10 and antioxidant enzymes were enhanced in OTH group. AP variability and baroreflex sensitivity were correlated with systolic AP, RPP, LF/HF ratio and inflammatory and oxidative stress parameters.

## Conclusion

The combination of HCTZ plus concurrent exercise training induced additional positive adaptations in cardiovascular autonomic control, inflammation and redox balance in ovariectomized SHR. Therefore, combining exercise and medication may represent a promising strategy for managing classic and remaining cardiovascular risks in AH.

## Introduction

Arterial hypertension (AH) is a chronic, complex and multifactorial disease [1] that can help to a disabling outcome if not well treated. The prevalence of AH increases throughout life and affects both sexes, but it is significantly higher in females after menopause. It is estimated that half of hypertensive women who are treated do not have AH under control [2]. Strategies to better treat AH has been intensively adopted over the decades and has impacted on a reduction in cardiovascular mortality in the last years [3]. However, the proper long-term control of the arterial pressure (AP) is still challenging [2, 4].

It is recognized the interplay between autonomic nervous system, inflammation, oxidative stress, and AH. Inflammation and oxidative stress exert joint effects on the central nervous system, increasing sympathetic activity. This activity, in turn, contributes to tissue inflammation, resulting in increased AP [5]. Clinical studies has demonstrated increase in pro-inflammatory cytokines, such as tumor necrosis factor alpha (TNF-α) [6] and interleukin 6 (IL-6) [7], with the severity of AH. In addition, we [8] and other authors [9] have reported elevated TNF-α [8, 9] and oxidative stress [8] in an experimental model of AH. On the other hand, interleukin 10 (IL-10), an anti-inflammatory cytokine, has been suggested to counterbalance the pro-inflammatory effects. Importantly, the IL-10/TNF-α ratio has been reported as a physiological parameter in mediating oxidative-stress-induced cardiac injury [10]. Moreover, we have previously demonstrated that ovarian hormone deprivation exacerbates the deleterious effects in the experimental AH [8].

The main goal of antihypertensive therapy is to keep AP well controlled. Diuretic are a common and effective therapy for AH, and they have been shown to reduce cardiovascular events [11]. The hydrochlorothiazide (HCTZ), a thiazide diuretic, has been one of the antihypertensive classes considered preferred in monotherapy for AP control [12], being widely used in clinical practice. Diuretic-induced natriuresis promote hypotensive effect, in which there is a reduction in the reabsorption of sodium chloride in the renal distal tubules. However, the decrease in peripheral vascular resistance appears to be determinant for long-term hypotension [13], corroborating with the potent vasodilatation effect induced by HCTZ on vascular smooth muscle cell [14].

Evidence suggests that long-term diuretic treatment has no positive effects on mechanisms related to the pathophysiology of AH. Chronic administration of diuretic could active

counter-regulatory mechanisms, thus inducing maintenance or even increase in sympathetic activity and in norepinephrine [15, 16], as well as, stimulating the renin-angiotensin-aldoste-rone system (RAAS) [17, 18]. Additionally, data suggested that thiazide diuretic have a diabe-togenic potential by aggravate metabolic dysfunction [19]. In this sense, even with a controlled AP, there is remaining cardiovascular risk in hypertensive patients.

On the other hand, the use of exercise training as a non-pharmacological approach is docu-mented and strongly recommended to management of the AH [12, 20]. Recent meta-analysis has showed that the combination of aerobic and resistance exercises performed in close peri-ods, termed concurrent exercise training (CET), shown similar potential to aerobic exercise in reduce AP [21]. Importantly, exercise training was associated with adaptations in key AP con-trol mechanisms such as autonomic, humoral, inflammation, and oxidative stress [8, 22–24]. In this sense, we previously reported an improvement in baroreflex sensitivity after exercise training [8, 23] related with increased sensitivity of afferent and/or efferent pathways [25] and vascular adaptations [26]. Moreover, exercise training is widely recognized to induce a sym-pathoinhibitory effect in AH in clinical and experimental studies [8, 27]. In addition, the observed increase in autonomic modulation in trained female SHR has been reflected in an improvement in inflammatory and in the oxidative stress profile, contributing to reduction in target organ damage [8, 22]. However, despite this evidence, the effects of the CET in associa-tion with HCTZ in the condition of menopause and hypertension are unknown.

Therefore, considering that counter-regulatory mechanisms act to restore AP levels prior to drug therapy [15–18], we investigated whether the association of exercise training during HCTZ treatment could play a role in the modulation of cardiovascular and autonomic mecha-nisms, as well as inflammatory and oxidative stress markers in relation to monotherapy alone in hypertensive female rats subjected to ovarian hormone deprivation. In this sense, in addi-tion to a better adjustment of mechanisms, the combination of approaches in AH could result in reduction in the long-term remaining risk.

## Materials and methods

Female *spontaneously hypertensive rats* (SHR) (150–200g, 90 days old) were obtained from Nove de Julho University (UNINOVE) (Sao Paulo, Brazil) and randomly allocated into 5 hypertensive groups (n = 7–8 rats each group): control (C), ovariectomized (O), ovariecto-mized treated with HCTZ (OH), ovariectomized submitted to exercise training (OT) and ovariectomized submitted to exercise training and treated with HCTZ (OTH). The rats were housed in cages (2–4 animals) in a temperature-controlled room (22–25°C) under a 12–hour dark/light cycle. Chow and water were offered ad libitum. The rat that did not complete a spe-cific evaluation was excluded from the statistical analysis. Ethics Committee on the Use of Ani-mals (CEUA) of Federal University of Sao Paulo (UNIFESP) (protocol n° 7611290618) approved the study and all surgical procedures and protocols were conducted in according with the recommendations.

### Ovariectomy

The rats were ovariectomized at 90 days of age as described previously [28]. Only the C group was not subjected to ovariectomy. The detailed procedure can be found in the S1 File.

### Pharmacological treatment

Pharmacological treatment was performed using HCTZ (Sanofi Medley Farmacêutica, Campi-nas, SP, Brazil), an antihypertensive drug corresponding to thiazide diuretics class [12], at a dose of 30 mg/kg/day. According to our pilot study conducted previously, this dose during 1

week showed sufficient to promote an AP reduction of approximately 10–12 mmHg in ovariectomized SHR.

The dose of HCTZ chosen in our study was selected with the aim of promoting AP reductions similar to the proposed exercise training. Thus, the pairing of the hypotensive effect between the approaches allows us to accurately assess which mechanisms (among those we evaluated) could be more benefited by HCTZ, exercise training and/or the combination of approaches.

The HCTZ tablet was macerated, diluted in drinking (filtered) water, and then was made available for consumption. Started on the same day as the first indirect AP measurement (i.e., one week after ovariectomy), all groups that would be treated with HCTZ underwent a 7–day adaptation period with the drug. Filtered water was consumed by the other groups that did not undergo pharmacological treatment during the entire adaptation period, as well as during the entire experimental protocol. Pharmacological treatment was continued for 8 weeks after the adaptation period. The daily consumption was monitored and then considered to adjust the amount of water for the groups undergo to the drug treatment.

### Tail-cuff plethysmography

All rats underwent tail plethysmography measurement in 4 moments of the study: baseline, after 7 days of medication adaptation (beginning of the intervention period), and at fourth and eighth weeks of the intervention period. Twenty consecutive and uninterrupted measurements were performed; during each assessment, with an interval of 15 seconds between each measurement, systolic arterial pressure (SAP) values were recorded (model BP-2000 Blood Pressure Analysis System, Visitech Systems, Inc, North Carolina, USA). The complete procedure is described in S1 File.

### Concurrent exercise training protocol

CET was performed on a motorized treadmill (aerobic exercise) and ladder adapted for rats (resistance exercise) for 3 days a week, on alternate days, for 8 weeks. Initially, all animals were adapted to the aerobic (0.3 km/h, 10 minutes) and resistance exercises (3 climbs without external overload) for 3 – 5 consecutive days. After that, they were submitted to a maximal running and maximal load tests, on different days, for the prescription of exercise training and determination of the maximal capacity of all groups. The details of this protocol are available in S1 File.

### Hemodynamic and cardiac functional assessment

On the last day of the protocol, rats were anesthetized and two polyethylene-tipped Tygon cannulas were implanted: into the carotid artery toward the left ventricle for direct AP recording and in the jugular vein for drug infusion, respectively.

Hemodynamic measurements were taken in conscious and awake rats in their cages, at least 24 hours after catheter placement. The arterial cannula was connected to a transducer (Blood Pressure XDCR, Kent Scientific), and AP signals were recorded over a 30-minutes period using a microcomputer equipped with an analog-to-digital converter (Windaq, 2 kHz sampling frequency, Dataq Instruments). The recorded data were analyzed on a beat-to-beat basis to quantify changes in SAP, diastolic AP (DAP), mean AP (MAP), and heart rate (HR) [28].

The cardiac effort was evaluated by the rate-pressure product (RPP), multiplying the SAP (mmHg) and the HR (bpm). RPP is considered a good measure of the cardiac function and has important clinical implications [29].

All animals were allocated into individual cages and subjected the same conditions regarding HCTZ availability and duration of HCTZ exposure before and after hemodynamic assessments, remaining until euthanasia.

## Baroreflex sensitivity assessment

After baseline AP measurement, baroreflex sensitivity was assessed by using increasing doses of phenylephrine (0.5 to 2.0 g/mL, intravenous) and sodium nitroprusside (5 to 20 g/mL, intravenous) given as sequential bolus injections (0.1 mL) to produce AP rise and fall responses ranging from 5 to 40 millimeter of mercury (mmHg) each. An interval between doses was necessary for AP to return to baseline values. Peak increases or decreases in MAP after phenylephrine or sodium nitroprusside injection and the corresponding peak reflex changes in HR were recorded for each dose of the drug. Baroreflex sensitivity was assessed by a mean index relating changes in HR to changes in MAP, allowing a separate analysis of gain for reflex bradycardia and reflex tachycardia [28].

## Vasopressor systems blockade

After baroreflex sensitivity assessment, AP was recorded under basal conditions for 5 minutes and then a vasopressin V1 receptor antagonist (aAVP) ([β-Mercapto-β,β-cyclopentamethyle-nepropionyl, O-me-Tyr$^2$, Arg$^8$]-Vasopressin; 10 µg/kg; Sigma-Aldrich, St Louis, MO, USA) was injected intravenously and the AP and signals were recorded over a period of 15 minutes. After that, losartan, an angiotensin II (Ang II) AT1 receptor antagonist (losartan potassium; 10 mg/kg, Medley Pharmaceuticals, Campinas, SP, BR) was injected intravenously after a new 5-minutes interval recorded and then 15 minutes were registered. Finally, 5 minutes were recorded and hexamethonium, a sympathetic ganglion blocking drug (hexane-1,6-bis[tri-methylammonium bromide]; 20 mg/kg, Sigma-Aldrich, St Louis, MO, USA) was injected intravenously. AP and HR were again recorded for 15 minutes [30]. The 5 minutes-average of MAP recording (lower values) after injection of each drug was used to assess the responses to vasopressor systems blockade. The difference between the MAP before and after injection of each drug was the response of each system.

## Heart rate and arterial pressure variability

Time domain analysis consisted of calculating mean pulse interval (PI) and SAP, with PI variability and SAP variability as the standard deviation from its respective time series. The entire tachogram was visualized by plotting the PI over time, and the three most stable sequences of 5 uninterrupted minutes from the total period were chosen. We chose one sequence at the beginning, one sequence in the middle, and one sequence at the end of the 30 minutes of AP recording. The sequences were individually analyzed for HR variability (HRV) and AP variability (APV) and the mean value of the 3 sequences was calculated for each animal. Both HRV and AVP were analyzed in time and frequency domains (spectral analysis was used for frequency domain parameters) with the CardioSeries (version 2.4, CardioSeries Software, Sao Paulo University, Brazil) software. Spectral power for low-frequency (LF, 0.20–0.75 Hz) and high-frequency (HF, 0.75–4.0 Hz) bands were calculated.

## Anthropometry and tissue collection

The rats were weighted weekly during the protocol.

The rats were pre-anesthetized with ketamine one day after hemodynamic evaluations and were submitted to euthanasia by decapitation. Heart, visceral white adipose tissue (WAT), and

skeletal muscles (soleus and plantaris) were collected. These tissues were immediately removed after euthanasia, properly weighed and the heart was immediately frozen at -80°C for inflammatory and oxidative stress analyses.

## Inflammatory mediators

A commercially available ELISA kit (R&D Systems Inc.) was used to assessment of TNF-α, IL-6 and IL-10 levels in cardiac tissue. The procedure was performed in accordance with the manufacturer's instructions. ELISA was performed in 96-well polystyrene microplate with a specific monoclonal antibody coating. Absorbance was measured at 540 nm in a microplate reader.

## Oxidative stress assessment

Cardiac tissues were cut into small pieces, placed in ice-cold buffer, and homogenized in an ultra-Turrax blender with 1g of tissue per 5 mL of 120 mmol/L KCl and 30 nmol/L phosphate buffer, pH 7.4. Homogenates were centrifuged at 4000 rpm for 10 minutes at 4°C. The supernatant was stored in a freezer at −80°C. Protein was determined as described previously [31].

Our study evaluated oxidative stress damage by lipoperoxidation (thiobarbituric acid reactive substances (TBARS)) and by protein oxidation (by carbonyls), pro-oxidant by nicotinamide adenine dinucleotide phosphate (NADPH) oxidase activity and hydrogen peroxide ($H_2O_2$) concentration, and antioxidant profile by enzymes activities: superoxide dismutase (SOD), catalase (CAT) and glutathione peroxidase (GPx) and by ferric reducing antioxidant power (FRAP), as well as nitrites in cardiac tissue. The techniques are detailed in the S1 File.

## Statistical analysis

Data are expressed as mean ± standard deviation. Distribution and homogeneity of variances were assessed using Shapiro-Wilk and Levene's tests, respectively. One-way analysis of variance (ANOVA) was used to compare the groups, followed by the Tukey post hoc test when appropriate. The Games-Howell post hoc test was used for lack of homogeneity of variances. Repeated measures ANOVA was used for tail plethysmography and exercise tests, and Bonferroni's post hoc test was used when necessary. The relationship between autonomic and antioxidant enzymes with SAP and HRV were analyzed by Pearson correlation analysis. The $p < 0.05$ value was considered as statistically difference between groups. IBM SPSS Statistical software for Windows (Version 23.0) was used to analyze the data.

## Results

### Anthropometry

Considering the diuretic and metabolic effects of HCTZ, we monitored the body weight (BW) and WAT of the animals to assess possible differences in weight gain throughout the study. Additionally, we evaluated skeletal muscle weight to determine if the exercise training model could alter this parameter compared to the non-trained groups. At the start of the protocol (day 1) and during the pre-intervention period, all groups had similar BW (grams) (week 2). At the end of the protocol, after 8 weeks of intervention (week 10), all ovariectomized groups showed increased BW compared with C group. However, weight gain during the study was higher in rats treated with HCTZ alone (OH: +22% vs. C: +10%). Heart and skeletal muscle weights were similar between groups. The cardiac hypertrophy index (HW/BW: heart weight/bodyweight ratio) did not differ between groups. It's worth noting that the OH groups had more WAT than the C group. There was no difference for daily feed intake. Water intake was

**Table 1. Anthropometric measurements and water and feed intake.**

|  | C | O | OH | OT | OTH | P |
|---|---|---|---|---|---|---|
| Initial BW, g | 179.3 ± 8.7 | 183.8 ± 10.4 | 180.8 ± 7.3 | 183.0 ± 10.5 | 182.5 ± 10.8 | 0.885 |
| Pre-intervention BW, g | 186.8 ± 11.7 | 198.3 ± 13.7 | 188.0 ± 8.6 | 200.8 ± 11.4 | 189.5 ± 12.5 | 0.073 |
| Final BW, g | 206.8 ± 11.1 | 233.5 ± 13.9* | 230.0 ± 9.0* | 233.3 ± 7.1* | 220.0 ± 9.2 | <0.001 |
| BW gain, g | 20.0 ± 11.1 | 35.3 ± 7.6 | 42.0 ± 5.8* | 32.5 ± 13.6 | 30.5 ± 13.0 | 0.005 |
| Heart weight, g | 0.800 ± 0.047 | 0.793 ± 0.052 | 0.780 ± 0.086 | 0.856 ± 0.119 | 0.860 ± 0.163 | 0.395 |
| HW/BW, mg/g | 3.87 ± 0.18 | 3.40 ± 0.17 | 3.39 ± 0.29 | 3.68 ± 0.57 | 3.91 ± 0.75 | 0.066 |
| WAT weight, mg | 0.625 ± 0.223 | 0.830 ± 0.185 | 0.906 ± 0.200* | 0.816 ± 0.147 | 0.658 ± 0.153 | 0.019 |
| Soleus weight, mg | 0.097 ± 0.009 | 0.100 ± 0.017 | 0.108 ± 0.007 | 0.102 ± 0.009 | 0.103 ± 0.018 | 0.190 |
| Plantaris weight, mg | 0.194 ± 0.014 | 0.210 ± 0.020 | 0.203 ± 0.046 | 0.215 ± 0.020 | 0.206 ± 0.019 | 0.597 |
| Daily feed intake, g | 15.9 ± 1.37 | 16.1 ± 2.10 | 16.8 ± 2.49 | 16.2 ± 1.59 | 16.4 ± 1.74 | 0.664 |
| Daily water intake, ml | 26.3 ± 3.19 | 32.0 ± 4.67* | 26.6 ± 3.91† | 30.3 ± 3.74*‡ | 24.9 ± 3.01†§ | <0.001 |

Data are presented as mean ± standard deviation (n = 8/group) and were analyzed using 1-way ANOVA followed by Tukey as a post hoc test.

Hypertensive control (C) and hypertensive ovariectomized rats: sedentary (O), treated with HCTZ (OH), trained (OT) and trained and treated with HCTZ (OTH).

* p < 0.05 vs. C

† p < 0.05 vs. O

‡ p < 0.05 vs. OH

§ p < 0.05 vs.

OT. BW, body weight; HW/BW, heart weight to body weight ratio; WAT, white adipose tissue.

higher in the O and OT groups compared with other groups. These findings can be found in the Table 1.

## Maximal exercise tests

Performance in exercises tests is an important indicator of the effectiveness of exercise training. We evaluated functional performance in both tests corresponding to the two types of proposed exercises (aerobic and resistance). For the maximal running test, there were main effects for moment (p < 0.001) (showing difference between evaluated moments), group (p < 0.021) (showing differences between studied groups) and moment-by-group interaction (p < 0.001) (showing a different effect of intervention (s) over time). Running performance increased in both trained groups in the fourth and eighth weeks when compared to their initial tests. In contrast, no changes in running performance were observed in non-trained groups during the intervention period. Performance in treadmill test was similar between groups before of the intervention protocol (C: 23.7 ± 1.4, O: 22.4 ± 1.5, OH: 22.4 ± 1.7, OT: 21.2 ± 2.1 and OTH: 21.0 ± 2.3 min). However, HCTZ plus CET group (OTH: 27.0 ± 1.8 min) presented a higher performance in relation to O (22.8 ± 1.6 min) and OH groups (22.2 ± 1.4 min) at the fourth week. In addition, performance in both trained groups (OT and OTH) were higher in relation C and OH groups at the final of the intervention (OT: 26.9 ± 3.7 and OTH: 26.8 ± 4.1 vs. C: 22.3 ± 2.0 and OH: 22.2 ± 2.3 min) (Fig 1A).

Maximal load test also presented a main effect for moment (p < 0.001), group (p < 0.001) and moment-by-group interaction (p < 0.001). The C group showed higher load performance in the final protocol compared with their respective baseline test. The HCTZ treatment alone induced increase in load performance at the fourth week, however this result was not maintained in the final test. Both trained groups showed an improvement in load test performance at the end of the intervention period (eight vs. first and fourth weeks). Similar performance was observed in the ladder test at the beginning of the protocol (C: 167 ± 19, O: 150 ± 16, OH:

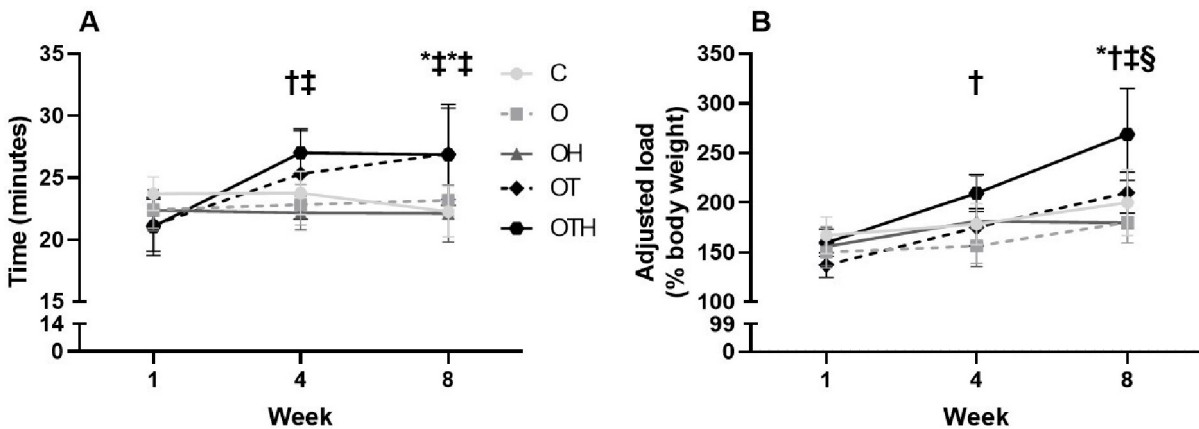

**Fig 1. Exercise performance tests in the studied groups.** (A) maximal running test (treadmill) and (B) maximal load test (ladder) in hypertensive control (C) and hypertensive ovariectomized rats: sedentary (O), treated with HCTZ (OH), trained (OT) and trained and treated with HCTZ (OTH). Data are presented as mean ± standard deviation (n = 8/group) and were analyzed using ANOVA for repeated measures with Bonferroni's correction, followed by Tukey as a post hoc test. * $p < 0.05$ vs. C; † $p < 0.05$ vs. O; ‡ $p < 0.05$ vs. OH, § $p < 0.05$ vs. OT.

156 ± 20, OT: 137 ± 13 and OTH: 159 ± 14% body weight). However, the combination HCTZ and CET induced a higher performance in relation to O group at the fourth week (OTH: 209 ± 19 vs. O: 156 ± 18% BW) and in relation to other groups at the eighth week (OTH: 269 ± 19 vs. C: 200 ± 33, O: 180 ± 21, OH: 179 ± 20 and OT: 210 ± 21% BW) (Fig 1B).

## Tail plethysmography

The tail plethysmography measurement allowed us to assess the AP behavior throughout the study, as well as confirm the effectiveness of the pharmacological treatment in controlling the AP of the groups treated with HCTZ. SAP measured using tail plethysmography showed group ($p < 0.001$) and moment-by-group interaction effects ($p = 0.001$). Similar SAP was observed between groups at the beginning of the protocol (adaptation phase) (C: 168 ± 3, O: 172 ± 7, OH: 172 ± 8, OT: 167 ± 6 and OTH: 164 ± 7 mmHg). The O group (O: 179 ± 8 mmHg) presented increase in SAP at the beginning of the intervention (post drug adaptation/ day 1 of the intervention period) in relation to C (164 ± 2 mmHg), OH (164 ± 8 mmHg) and OTH groups (155 ± 7 mmHg). Higher SAP values were observed in O group again in the fourth (C: 166 ± 10; OH: 167 ± 9, OTH: 159 ± 11 vs. O: 187 ± 5 mmHg) and at the end of the intervention period (C: 165 ± 9; OH: 169 ± 6 and OTH: 161 ± 12 vs. O: 185 ± 12 mmHg). In addition, the OT group showed lower SAP in relation to O group also at the fourth (OT: 163 ± 14 mmHg) and eighth weeks (OT: 168 ± 12 mmHg) (see S1 Fig).

## Hemodynamic and cardiac functional assessment

The final hemodynamic assessment provides information on the effectiveness of the antihypertensive treatments (medication and exercise training) in the ovariectomized groups that received the intervention either individually or in combination. The rats submitted to intervention (HCTZ treatment alone, exercise training alone or associated with HCTZ) showed lower SAP ($p = 0.006$) (OH: 181 ± 15, OT: 178 ± 18 and OTH: 178 ± 18 mmHg) compared to O group (206 ± 15 mmHg), but there were no differences when compared with the C group (196 ± 20 mmHg). Furthermore, CET alone or combined with HCTZ promoted a lower MAP ($p = 0.013$) in relation to O group (OT: 153 ± 15 and OTH: 153 ± 17 vs. O: 178 ± 17 mmHg); however, without difference in relation to other groups (C: 167 ± 16 and OH: 156 ± 13

mmHg). No differences were observed between groups for DAP (C: 141 ± 15, O: 152 ± 21, OH: 133 ± 13, OT: 130 ± 13 and OTH: 131 ± 17 mmHg, p = 0.054). In turn, resting bradycardia was observed in trained groups (p = 0.002) (OT: 331 ± 21 and OTH: 346 ± 24 bpm) in relation to C (384 ± 28 bpm vs. OT) and O groups (387 ± 36 bpm vs. OT and OTH), but without differences in relation to OH group (366 ± 30 bpm) after 8 weeks of intervention. The groups that received pharmacological treatment and/or underwent exercise training showed lower RPP (mmHg*bpm x $10^3$) compared to the ovariectomized group (OH: 66.5 ± 8.3, OT: 59.0 ± 8.0 and OTH: 61.3 ± 6.0 vs. O: 80.1 ± 11.6, p < 0.001). The RPP was also reduced in trained groups (OT and OTH) in relation to the C group (75.1 ± 9.5) (Fig 2A–2E).

## Baroreflex sensitivity

Considering that the baroreceptor reflex is impaired in AH, we evaluated whether HCTZ, exercise training, or the combination of both could restore this important reflex by increasing the sensitivity of the bradycardic and tachycardic responses. Both trained groups (OT and OTH) presented an increased bradycardic response to phenylephrine compared with O group (OT: −1.3 ± 0.4 and OTH: −1.6 ± 0.3 vs. O: −0.6 ± 0.3 bpm/mmHg) (p < 0.001). The C (−1.0 ± 0.3 bpm/mmHg) and OH groups (−1.1 ± 0.6 bpm/mmHg) did not show differences in bradycardic response compared with other groups. Moreover, the O group showed a decrease in tachycardic response (bpm/mmHg) to sodium nitroprusside compared with C group (C: −2.4 ± 0.4 vs. O: −1.3 ± 0.5 bpm/mmHg). However, trained alone or combined with HCTZ groups showed an increase in tachycardic response compared with O group (OT: −2.4 ± 0.8 and OTH: −2.4 ± 0.8 vs. O: −1.3 ± 0.5 bpm/mmHg) (p = 0.004) (Fig 3A and 3B).

## Heart rate variability and arterial pressure variability

In our protocol, measures of HRV and APV provide information about cardiac and vascular autonomic modulation, respectively. In this regard, elevated sympathetic modulation of both cardiac and vascular systems is observed in AH. We did not observe differences between the groups in terms of PI variance (Var-PI) and root mean square of successive differences between normal heartbeats (RMSSD) between groups. However, we found that HCTZ plus CET (OTH group) induced a lower LF band and a higher HF band of PI compared to the O group (Table 2), resulting in a lower sympathovagal balance (LF/HF ratio).

Both trained groups (OT and OTH) presented lower variance of SAP (Var-SAP) compared with C and O. Regarding vascular sympathetic modulation, OT and OTH groups presented reduction in the LF band of SAP (LF-SAP) compared with non-trained groups (Table 2).

## Vasopressor systems blockade

Sequential blockade of vasoconstrictor mechanisms allows investigation of the contribution of each evaluated mechanism to the basal AP levels, which is of interest in a condition of AH and postmenopause undergoing antihypertensive approaches. There were no differences in MAP (mmHg) after aAVP and Ang II AT1 receptor blockade. However, hexamethonium injection resulted in a greater reduction of MAP in the O group when compared to the C group. Furthermore, the reduction in MAP after autonomic ganglia blockade was lower in the OTH group compared to the O group (Table 3).

## Inflammatory mediators

Inflammation is directly involved in the pathophysiology of AH. In our study, we evaluated important pro-inflammatory cytokines (TNF-α and IL-6) involved in this condition, as well as

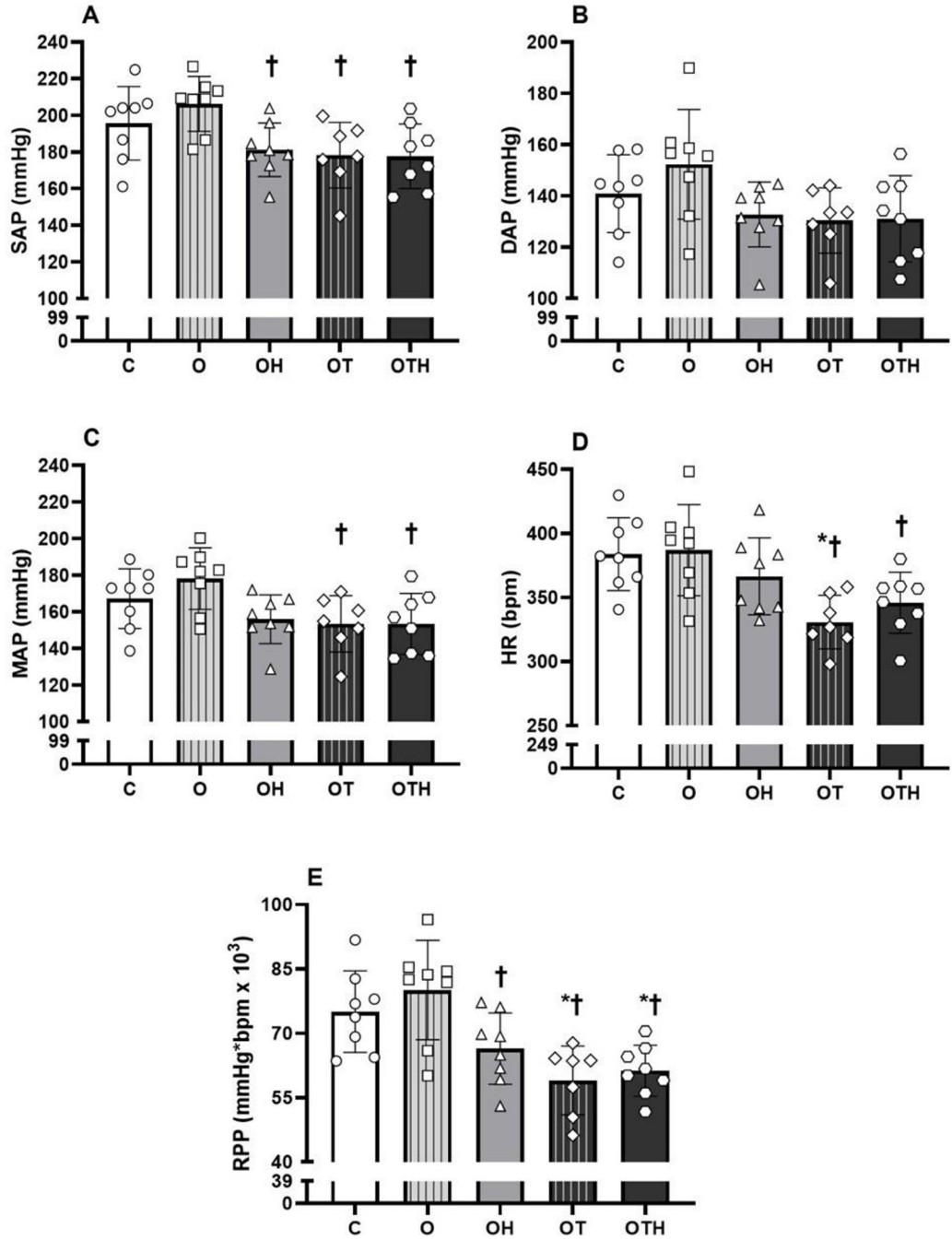

**Fig 2. Hemodynamic and cardiac function assessments.** (A) Systolic, (B) diastolic and (C) mean arterial pressure, (D) resting heart rate and (E) rate-pressure product in hypertensive control (C) and hypertensive ovariectomized rats: sedentary (O), treated with HCTZ (OH), trained (OT) and trained and treated with HCTZ (OTH). Data are presented as mean ± standard deviation (n = 7-8/group) and were analyzed using 1-way ANOVA followed by Tukey as a post hoc test. * $p < 0.05$ vs. C; † $p < 0.05$ vs. O; ‡ $p < 0.05$ vs. OH. SAP, systolic arterial pressure; DAP, diastolic arterial pressure, MAP, mean arterial pressure; HR, heart rate; RPP, rate-pressure product.

the potential of the approaches to induce possible increases in IL-10, an anti-inflammatory cytokine. IL-6 levels in cardiac tissue were similar between groups. However, OTH group presented higher levels of cardiac TNF-α (vs. C) and IL-10 (vs. O and OH groups). OT showed

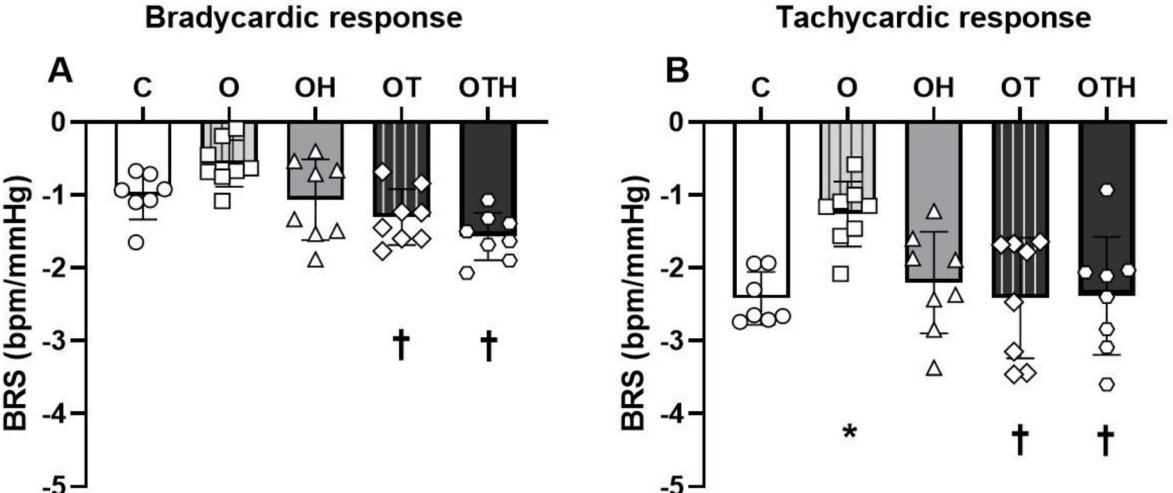

**Fig 3. Baroreflex sensitivity assessed by administration of increasing doses of phenylephrine and sodium nitroprusside.** (A) Bradycardic and (B) tachycardic response of the baroreflex sensitivity in hypertensive control (C) and hypertensive ovariectomized rats: sedentary (O), treated with HCTZ (OH), trained (OT) and trained and treated with HCTZ (OTH). Data are presented as mean ± standard deviation (n = 7-8/group) and were analyzed using 1-way ANOVA followed by Tukey as a post hoc test. * $p < 0.05$ vs. C; † $p < 0.05$ vs. O. BRS, baroreflex sensitivity.

higher IL-10/TNF-α ratio compared with non-trained groups, while the combination with HCTZ result in increased ratio in relation all groups (Table 4).

## Oxidative stress

Similar to the inflammation, oxidative stress plays a key role in the development and progression of AH. In this regard, we aimed to investigate parameters of oxidative damage, as well as pro-oxidant and antioxidant profile, focusing on the effects of the approaches used in the study. Protein oxidation was increased after ovarian hormone deprivation (OS group) (4.62 ± 0.85 nmol/mg protein) in relation to all groups (C: 3.69 ± 0.75, OH: 3.87 ± 0.48, OT:

**Table 2. Heart rate variability and arterial pressure variability.**

| | C | O | OH | OT | OTH | p |
|---|---|---|---|---|---|---|
| Var-PI, ms$^2$ | 56.2 ± 18.2 | 57.8 ± 21.8 | 53.5 ± 19.4 | 45.7 ± 18.1 | 60.0 ± 20.1 | 0.515 |
| RMSSD, ms | 6.6 ± 1.0 | 6.1 ± 0.7 | 5.8 ± 0.9 | 5.9 ± 1.6 | 6.6 ± 1.1 | 0.488 |
| LF, nu | 22.9 ± 5.4 | 30.2 ± 4.0 | 24.0 ± 6.7 | 25.7 ± 4.5 | 22.2 ± 4.9† | 0.043 |
| HF, nu | 77.1 ± 5.4 | 69.8 ± 4.0 | 76.0 ± 6.7 | 74.3 ± 4.5 | 77.8 ± 4.9† | 0.043 |
| LF/HF | 0.32 ± 0.10 | 0.45 ± 0.09 | 0.34 ± 0.10 | 0.37 ± 0.09 | 0.30 ± 0.08† | 0.036 |
| Var-SAP, mmHg$^2$ | 53.3 ± 13.6 | 60.1 ± 13.1 | 44.7 ± 10.5 | 33.5 ± 13.2*† | 30.3 ± 10.5*† | <0.001 |
| LF-SAP, mmHg$^2$ | 16.3 ± 6.7 | 18.8 ± 6.1 | 15.8 ± 5.5 | 6.9 ± 6.0*†‡ | 6.2 ± 5.2*†‡ | <0.001 |

Data are presented as mean ± standard deviation (n = 7-8/group) and were analyzed using 1-way ANOVA followed by Tukey as a post hoc test.

Hypertensive control (C) and hypertensive ovariectomized rats: sedentary (O), treated with HCTZ (OH), trained (OT) and trained and treated with HCTZ (OTH).

* $p < 0.05$ vs. C

† $p < 0.05$ vs. O

‡ $p < 0.05$ vs.

OH. Var-PI, variance of pulse interval; RMSSD, root mean square of successive differences between normal heartbeats; LF, low frequency band; HF, high frequency band; Var-SAP, variance of systolic arterial pressure; LF-SAP, low frequency band of systolic arterial pressure.

**Table 3. Reduction in mean arterial pressure after vasopressor systems blockade.**

| Δ MAP, mmHg | C | O | OH | OT | OTH | p |
|---|---|---|---|---|---|---|
| aAVP | -3.3 ± 1.8 | -6.9 ± 4.9 | -6.3 ± 3.5 | -4.4 ± 1.8 | -5.0 ± 1.6 | 0.178 |
| Losartan | -9.8 ± 3.8 | -12.5 ± 3.0 | -8.7 ± 2.1 | -9.5 ± 6.0 | -8.7 ± 4.0 | 0.381 |
| Hexamethonium | -53.2 ± 12.6 | -75.4 ± 18.4* | -54.9 ± 9.0 | -62.9 ± 15.7 | -49.8 ± 12.4† | 0.009 |

Data are presented as mean ± standard deviation (n = 7-8/group) and were analyzed using 1-way ANOVA without (hexamethonium) or with Welch's correction (aAVP and losartan), followed by Tukey (hexamethonium) as a post hoc test.

Hypertensive control (C) and hypertensive ovariectomized rats: sedentary (O), treated with HCTZ (OH), trained (OT) and trained and treated with HCTZ (OTH).

* p < 0.05 vs. C

† p < 0.05 vs.

O. aAVP, vasopressin V1 receptor antagonist.

3.48 ± 0.44 and OTH: 3.69 ± 0.49 nmol/mg protein, p = 0.010). Pro-oxidant profile (NADPH oxidase) was higher in O in relation to C group (O: 0.39 ± 0.12 vs. C: 0.24 ± 0.07 nmol/mg protein), without differences to OH group (OH: 0.31 ± 0.07). On the other hand, trained groups showed decrease in NADPH oxidase in relation to O group (OT: 0.27 ± 0.06 and OTH: 0.24 ± 0.09 nmol/mg protein vs. O) (p = 0.005). In addition, OH, OT and OTH groups showed reduced cardiac $H_2O_2$ levels in relation to C group (OH: 1.7 ± 0.3, OT: 2.0 ± 0.4 and OTH: 1.8 ± 0.4 vs. C: 2.7 ± 0.6 μM/mg protein, p < 0.001). Interestingly, C and OTH groups showed increased cardiac SOD activity compared to OH and OT groups (C: 14.0 ± 1.5 and OTH: 13.7 ± 0.6 vs. OH: 11.9 ± 1.0 and OT: 11.7 ± 0.3 USOD/mg protein, p < 0.001). However, any differences were observed between O and other groups (12.4 ± 0.8 USOD/mg protein). The O group presented a lower cardiac CAT activity in relation to C group (O: 0.87 ± 0.28 vs. C: 1.35 ± 0.24 nmol/mg protein), without differences in relation to OH and OT (OH: 1.16 ± 0.14 and OT: 1.07 ± 0.21 nmol/mg protein). However, OTH group reduce this parameter (OTH: 1.20 ± 0.17 nmol/mg protein vs. O). Finally, HCTZ combined with CET promote increase in cardiac GPx (nmol/mg protein) in relation to O group (OTH: 27.1 ± 3.24 vs. O: 19.4 ± 5.71 nmol/mg protein, p = 0.024), but there were no statistical differences compared with C (24.4 ± 5.86 nmol/mg protein), OH (24.0 ± 3.34 nmol/mg protein) and OT groups (24.4 ± 2.97 nmol/mg protein) (Fig 4A–4F). Oxidative stress induced-cardiac damage evaluated by TBARS (C: 2.95 ± 0.72, O: 3.52 ± 1.09, OH: 2.37 ± 0.88, OT: 3.03 ± 0.95 and OTH: 3.47 ± 1.59 μmol/mg protein, p = 0.224), FRAP (C: 0.46 ± 0.22, O: 0.46 ± 0.20, OH: 0.50 ± 0.19, OT: 0.61 ± 0.14

**Table 4. Inflammatory mediators assessed in cardiac tissue.**

| | C | O | OH | OT | OTH | p |
|---|---|---|---|---|---|---|
| TNF-α, pg/mg protein | 42.8 ± 8.0 | 46.7 ± 16.7 | 41.5 ± 10.3 | 33.9 ± 7.0 | 31.8 ± 5.3* | 0.025 |
| IL-6, pg/mg protein | 85.3 ± 14.9 | 87.9 ± 33.4 | 90.3 ± 15.2 | 93.2 ± 13.3 | 99.6 ± 14.8 | 0.488 |
| IL-10, pg/mg protein | 65.2 ± 12.5 | 52.8 ± 23.2 | 45.1 ± 10.6 | 59.9 ± 10.4 | 78.1 ± 17.3†‡ | 0.004 |
| IL-10/TNF-α | 1.32 ± 0.21 | 1.18 ± 0.12 | 1.19 ± 0.29 | 1.72 ± 0.19*†‡ | 1.99 ± 0.07*†‡§ | <0.001 |

Data are presented as mean ± standard deviation (n = 7-8/group) and were analyzed using 1-way ANOVA without (IL-6 and IL-10) or with Welch's correction (TNF-α), followed by Tukey (IL-10 and IL-10/TNF-α) and Games-Howell (TNF-α) as a post hoc test.

Hypertensive control (C) and hypertensive ovariectomized rats: sedentary (O), treated with HCTZ (OH), trained (OT) and trained and treated with HCTZ (OTH).

* p < 0.05 vs. C

† p < 0.05 vs. O

‡ p < 0.05 vs. OH

§ p < 0.05 vs. OT.

TNF-α, tumor necrosis factor alpha; IL-6, interleukin 6; IL-10, interleukin 10.

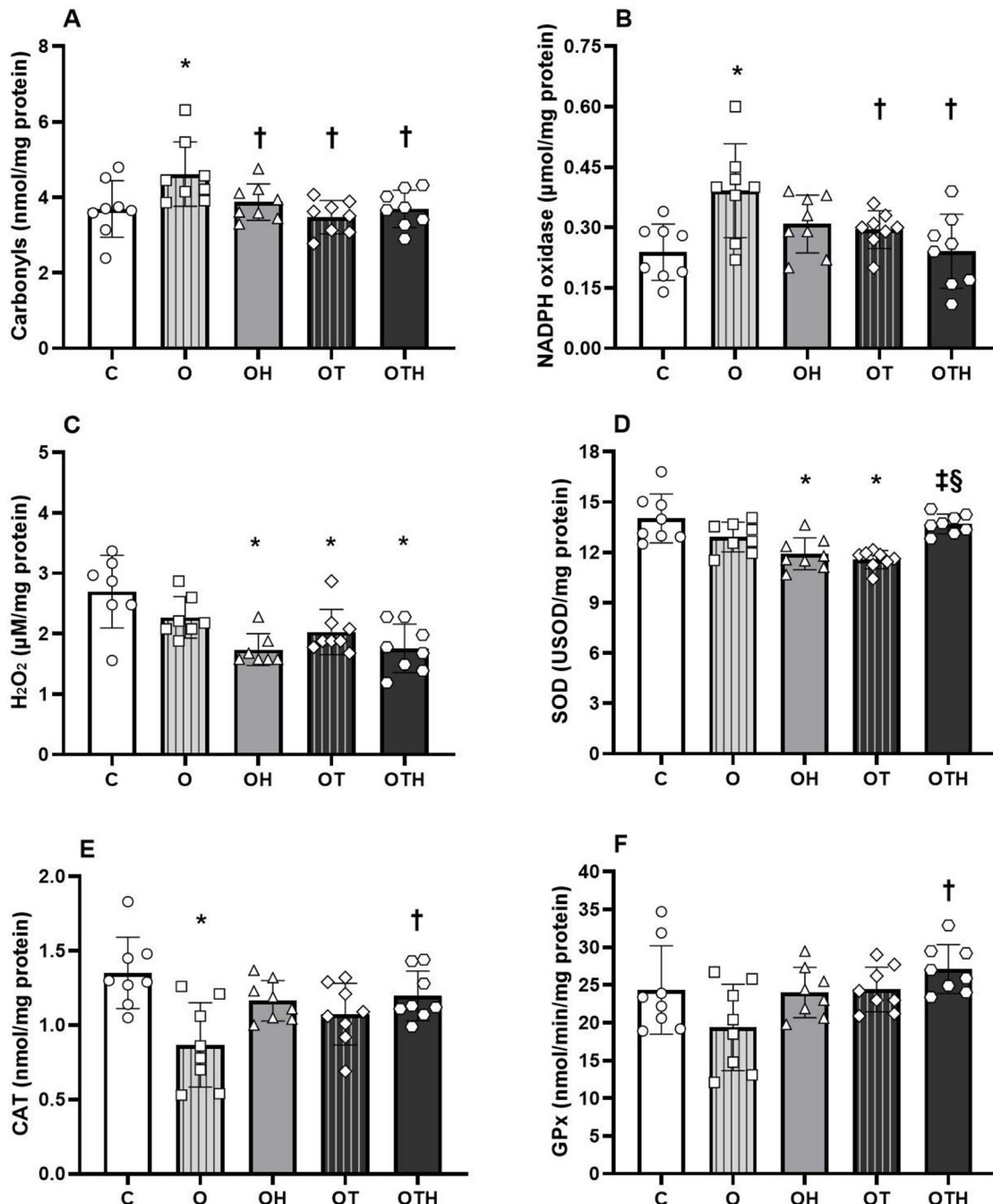

**Fig 4. Oxidative stress assessed in cardiac tissue.** Oxidative stress damage (A), pro-oxidant (B and C) and antioxidant enzymes (D–F) assessed in hypertensive control (C) and hypertensive ovariectomized rats: sedentary (O), treated with HCTZ (OH), trained (OT) and trained and treated with HCTZ (OTH). Data are presented as mean ± standard deviation (n = 7-8/group) and were analyzed using 1-way ANOVA followed by Tukey as a post hoc test. * $p < 0.05$ vs. C; † $p < 0.05$ vs. O; ‡ $p < 0.05$ vs. OH; § $p < 0.05$ vs. OT. NADPH, nicotinamide adenine dinucleotide phosphate; $H_2O_2$, hydrogen peroxide; SOD, superoxide dismutase; CAT, catalase; GPx, glutathione peroxidase.

and OTH: 0.54 ± 0.11 mM Fe(ii), p = 0.424) and nitrite concentrations (C: 1.67 ± 0.35, O: 1.70 ± 0.20, OH: 1.66 ± 0.16, OT: 1.90 ± 0.31 and OTH: 1.82 ± 0.24 μmol/mg protein, p = 0.296) were not different between studied groups.

## Correlations

We investigate whether these parameters could be related to AP, autonomic and inflammatory changes in studied groups based on a significant improvement in APV in OT group. Our data showed positive relationships between Var-SAP and LF-SAP with SAP, RPP, LF/HF ratio and IL-10/TNF-α ratio. Strong relationship between Var-SAP with LF-SAP was observed (see S1 Table).

The improvement in bradycardic response of baroreflex sensitivity of SAP was correlated with reduced SAP (R = 0.55, p < 0.001), RPP (R = 0.64, p < 0.001), Var-SAP (R = 0.61, p < 0.001), TNF-α (R = 0.5, p = 0.001) and protein oxidation (R = 0.54) (p < 0.005). RPP showed negative relationship with IL-10/TNF-α ratio (R = −0.48, p = 0.004). Finally, the cardiac IL-10/TNF-α ratio was inversely correlated with cardiac protein oxidation (R = −0.54, p = 0.001).

## Discussion

The combined effect of medication and exercise is not simply the sum of their individual effects [32]. Therefore, we investigated the ability of CET associated with HCTZ in to modulate mechanisms related to AH pathophysiology in this study. We use a hypertensive model of postmenopause aimed at experimentally simulating the condition of postmenopausal females, a population in which AH is more prevalent and affected by cardiofunctional impairment. The main finding from this study is that an additional adjustment on AP control mechanisms is achieved when exercise training is associated with HCTZ treatment in hypertensive ovariectomized rats. The benefits are extended to functional capacity, cardiovascular and autonomic control, as well as cardiac anti-inflammatory and oxidative stress profiles. Thus, exercise practice during medication treatment could directly contribute to minimize the progression of the AH-related morbimortality.

Regarding metabolic results, previous study demonstrated that HCTZ has been linked to metabolic disorders, such as insulin resistance and fat storage [19]. In our study, we did not measure glycemic or insulin levels. However, we found that the non-exercised HCTZ-treated group gained more BW and had more WAT weight (see Table 1). In contrast, BW gain and WAT were similar in control and trained groups, suggesting that the medication approach can induce anthropometric changes even in the absence of increased feed intake. We believe that the participation of the RAAS may be one of the possible mechanisms related to the anthropometric effect, since the literature has shown an increase in the RAAS after HCTZ [18]. In fact, the production of angiotensinogen by adipose tissue promotes adipogenesis, and Ang II can induce the process of adipose tissue accumulation by increasing the production of key lipogenic enzymes and inducing lipogenesis through the AT2 receptor pathway [33]. In contrast, the lack of greater BW gain in the trained group treated with HCTZ could be explained by down-regulation of vasoconstrictor axis of the RAAS and the likely increased energy expenditure induced by CET.

In addition to metabolic benefits, our data showed that the performance in the treadmill exercise test improved in both trained groups, as previously reported [8], but significantly improved in the resistance exercise test only when CET was combined with HCTZ. In fact, older hypertensive adults using 25–50 mg of HCTZ daily did not presented alterations in VO2 peak after treatment [34]. However, clinical evidence has showed an increase in the treadmill walking time after combination of HCTZ and amiloride in patient with chronic stable angina [35]. Furthermore, impaired muscle adaptation has been reported when aerobic and resistance exercise are including in a same exercise training program [36]. Therefore, the marked enhancement in maximal strength performance of the OTH group in the present study suggest

that HCTZ, when combined with exercise training, appears significantly improve strength production during maximal load test in hypertensive ovariectomized rats. However, we have no knowledge of evidence on how exercise training could affect the pharmacodynamics of HCTZ. Given that the magnitude enhance in the OTH group was also better than the OT group in the resistance exercise test, our data suggests that the higher performance in the OTH group may be related to a possible effect of HCTZ in trained rats rather than the opposite. On the other hand, we speculate that higher BW gain and WAT weight in the OH group may have hampered performance. The possibility of exercise manipulating the pharmacokinetic and pharmacodynamic effects of the HCTZ is a point to be investigated and further experimental evidence are needed to clarify this issue.

Regarding hemodynamic evaluations, HCTZ alone promoted SAP reduction in both indirectly and directly AP assessment in ovariectomized hypertensive rats. Furthermore, our CET protocol alone or combined with HCTZ was effective in reducing SAP and MAP. These data demonstrated that both therapies (OH, OT and OHT groups) were able to avoid the increase in SAP associated with ovarian hormones deprivation in hypertensive rats in the present study. It is important to emphasize that thiazide diuretics are among the preferred classes for antihypertensive treatment in monotherapy or in combination with other classes. The usual daily dose of HCTZ ranges between 25–50 mg [12]. However, inadequate AP control after medication treatment in AH is common and strategies such as increasing the dose or combination with other classes of antihypertensive are used. In fact, compensatory upregulation of the AP control mechanism has been implicated in response to HCTZ. In these sense, clinical [16, 17, 37] and experimental studies [18] have reported that sympathetic nervous system [16] and RAAS [17, 18, 37] are stimulated by HCTZ, including in low doses of HCTZ (e.g. 25 mg daily) [17]. On the other hand, exercise training has been demonstrated positively modulate autonomic nervous systems thought reduction in sympathetic activity [38]. Regarding the RAAS, evidences have shown that exercise training modulates both the Ang II/AT1 and Ang 1-7/Mas axes, promoting a down and up regulation, respectively [39, 40]. Water intake during the experimental protocol (see Table 1) was comparable between OH and OTH groups in our study. In this regard, CET shown a better potential to promote MAP control probably due to the high potential of this approach to improve cardiovascular autonomic control.

The baroreceptor reflex regulates moment-to-moment the AP and plays a key role for maintaining cardiovascular homeostasis. Exercise training has previously been shown to improve baroreflex sensitivity in female ovariectomized rats [8, 22, 28]. In the present study, we showed that ovariectomy reduced the tachycardic responses of baroreflex sensitivity. Despite HCTZ alone did not change baroreflex sensitivity, CET alone or combined with HCTZ improved both bradycardic and tachycardic responses of baroreflex sensitivity, suggesting that the improvement observed in OTH was likely determined by the potential effect of exercise and not the effect of HCTZ. The bradycardic response is more vagal-dependent, while the tachycardic responses is more sympathetic-dependent. The increase and decrease of the vagal and sympathetic activity, respectively, has been cumulatively supported as a positive adaptation to exercise training. A pertinent hypothesis that could to explain our data is the possibility of increased sensitivity of the afferent and/or efferent pathways, as well as the sino-atrial node to the heart, resulting in better reflex responses [25]. Moreover, evidence has suggested that a reduction in vascular wall distensibility and structural alterations in large arteries are associated with chronic baroreceptor dysfunction in AH [41]. On the other hand, improvements in carotid arterial distensibility index combined with improved baroreflex sensitivity after exercise have been observed in humans [26]. Experimentally, vascular adaptations such as vasodilation and vascular compliance have been shown to improve in chronically trained SHR [42]. Therefore, we hypothesized that the improvement in baroreflex sensitivity may be

related to neural component and mechanosensitive adaptations of the arteries induced by training.

We hypothesize that improvements in vagal modulation, as measured by the increase in HF band and indirectly assessed by HRV, contributed to improvements in bradycardic responses. Although our study found no improvement in vagal modulation in the OT group as measured by HRV, it is possible that CET alone increased vagal tonus to the heart [43], as evidenced by the expressive resting bradycardia seen in the OT group. In the same way, it is possible that reduction in cardiac, vascular and overall sympathetic contribution, as verified by the reduction of the LF (nu) (OTH group), LF-SAP (OT and OTH groups) and the lower reduction of the MAP after hexamethonium injection (OTH group), respectively, support the enhanced tachycardic responses.

Sympathetic overactivity is one of the most well documented physiological dysfunctions in AH-related genesis and end-organ damage [44]. Furthermore, postmenopausal women had a higher sympathetic contribution to AP levels than younger women [44]. Thus, strategies to reduce cardiovascular autonomic dysfunction in postmenopausal women are clinically desired. In the present study, no statistically significant effects were observed for HCTZ and exercise training on the contribution of vasopressin and RAAS system in the baseline MAP. Despite we did not measure RAAS components, there is some evidence indicate an increase after chronic HCTZ treatment [17, 18, 37]. In addition, Ang II can stimulate sympathetic activation [45], which may exacerbate this mechanism which is elevated in AH. Furthermore, evidence has reported a sympathoexcitatory effect promoted by diuretics [46]. However, the influence of the HCTZ on sympathetic activity are inconsistent. In this sense, plasma norepinephrine and norepinephrine release rate remained unchanged in elderly hypertensives after six months of treatment with HCTZ [47]. In obese hypertensive individuals, Grassi et al. [15] also did not observed differences in muscle sympathetic nerve activity after 12 week of HCTZ treatment. In the present study, ovariectomized group (O) showed greater sympathetic contribution on the basal MAP values than the C group after autonomic ganglia blockade by hexamethonium; and the HCTZ plus CET reduce the sympathetic tonus on MAP. This finding supports the positive role of CET in the management of the counter-regulatory mechanism induced by HCTZ treatment, most likely modulating inflammation and oxidative stress and resulting in improved AP control.

It is important to remind that high APV observed in AH is an important factor related to end-organ damage, such as in the cardiac and renal tissues [48]. In this sense, in comparison to RAAS inhibitors, a meta-analysis found an association between diuretics (chlorthalidone and HCTZ) and lower left ventricular mass [49]. In addition to SAP control, male SHR who received four months of HCTZ treatment alone [48] or in combination with nifedipine [50] showed reduction in APV, as well as an increase in baroreflex sensitivity and a reduction in end-organ damage, such as ventricle, kidney and aortae. We found no change in baroreflex sensitivity, Var-SAP or LF-SAP in ovariectomized SHR submitted to HCTZ treatment alone. These findings suggest that there may be sex differences, probably exacerbated by ovarian hormone deprivation, in long-term HCTZ treatment. Aside from sex, the duration of intervention may have contributed to the partial reproducibility of these results. Furthermore, in the present study only trained groups showed a significant reduction in APV when compared to the non-trained groups. Otherwise, our findings are consistent with current evidence, which has shown that a combination of aerobic and resistance exercise can reduce APV in hypertensive patients [50] and SHR [8]. More importantly, we showed that lower AP values in studied rats were associated with improvements in Var-SAP and LF-SAP, LF/HF ratio and baroreflex sensitivity, reinforcing the role and interaction of cardiovascular autonomic changes in AP management in this model of hypertension and postmenopausal. Moreover, we observed that the

improvement on bradycardic response of baroreflex sensitivity was correlated with better APV in studied animals.

Elevated levels of pro-inflammatory cytokines such as TNF-α [51, 52], IL-1 [51, 52] and IL-6 [51] regulate sympathetic flow and AP, while IL-10 [52] exerts beneficial effects on these parameters. Furthermore, pro-inflammatory cytokines leads to an increased generation of reactive oxygen species [53, 54]. In this context, TNF-α and oxidative stress levels were previously found to be elevated in ovariectomized SHR [8]. Fukuzawa et al. [55], on the other hand, reported that HCTZ had no effect on TNF-α production in vivo or in vitro cells, which is consistent with our findings. However, when HCTZ and CET were combined, cardiac IL-10 levels and IL-10/TNF-α ratio were significantly lower (vs. O group). In this sense, we previously shown decrease in anti-inflammatory cytokines after ovariectomy [8], and that aerobic and resistance exercise training improved renal (increasing IL-10) [8] and cardiac inflammation (reducing TNF-α and IL-6) in ovariectomized SHR [8, 24]. Reduced TNF-α after exercise training has been reported also in male SHR [9]. Moreover, decreased levels of SOD and CAT, as well as increased lipoperoxidation and reactive oxygen species, have been observed in cardiac myocytes after exposure to TNF-α. Importantly, treatment with IL-10 prevented all these changes [10]. The antioxidant effect of IL-10 under conditions of increased oxidative stress has been also documented. IL-10 treatment has been shown to improve SOD and CAT activities, as well as the redox ratio in renal ischemia-reperfusion induced by lipid peroxidation [56]. Recently, Qiu et al. [57] demonstrated that IL-10 reverses AH-induced vascular hypertrophy, possibly through its antioxidant and anti-inflammatory effects. Moreover, Kaur et al. [10] have highlighted the antioxidant properties of IL-10 by mitigating the antioxidant changes caused by TNF-α. Additionally, the authors suggest that IL-10 and TNF-α have significant physiological implications in clinical conditions [10]. Based on our data, it is plausible to speculate that exercise training contributed more to anti-inflammatory and antioxidant adaptations in the OTH group than HCTZ alone. Moreover, the anti-inflammatory effects and reported antioxidant effect promoted by IL-10 may have contributed to the improvement of the cardiac oxidative stress profile observed in the trained rats, mainly in the OTH group, probably positively impacting in the cardiac function (RPP) and in the functional capacity.

It is also worth noting that the sympathetic nervous system regulates the pro-inflammatory response [58]. Indeed, our findings showed a link between decreased LF-SAP and Var-SAP, and a lower cardiac IL-10/TNF-α ratio. As a result, we believe that exercise-induced effect on APV can positively modulate the cardiac inflammatory profile, resulting in less end-organ damage in AH.

Reactive oxygen species are involved in signaling from cell to system, and pro-oxidant enzymes are implicated in an oxidative profile [58]. In AH, oxidative stress is regarded as a common, but non-unique, factor that influences the local and systemic processes that favors AH [58]. In addition, inflammatory mediators are known to active pro-oxidants and influence redox balance. In this regard, the current study found that an increase in cardiac IL-10/TNF-α ratio was associated with lower protein oxidation. We also found high levels of pro-oxidant (NADPH oxidase) and protein oxidation, as well as a reduction in the antioxidant CAT activity in the cardiac tissue of the O group.

Additionally, in according with our data, HCTZ alone was not associated with amelioration of the vascular NADPH oxidase or superoxide anion levels [59] and did not change TBARS concentrations [60] in male SHR. However, in the present study 8-weeks of HCTZ treatment restored cardiac protein oxidation in ovariectomized SHR. Moreover, the combination of HCTZ with CET ameliorates pro-oxidant (NADPH oxidase and $H_2O_2$), enhanced antioxidants (CAT, SOD and GPx activities) in cardiac tissue, probably resulting in reduced protein oxidation, reflecting better redox balance in this group.

In fact, there is consistent evidence that antioxidant defense increase in response to aerobic or resistance exercises in cardiac and kidney tissues [22, 61]. Exercise training could induce activation of the nuclear factor erythroid 2-relatede factor 2 (Nrf2), which is a most important transcription factor recognized for regulating antioxidant response, acting on a specific portion of DNA, known as AREs (antioxidant response elements), encoding antioxidant enzymes, such as SOD, CAT, and GPx [62]. Furthermore, there are evidence that exercise training could promote a reduction in norepinephrine levels, resulting in an inhibitory effect on pro-inflammatory (TNF-α) and oxidative stress profile (reduction in NADPH oxidase and superoxide anion), reflecting in an improved cardiac function [63]. In the same way, it is important to remind that we observed in the present study reduced Var-SAP, LF-SAP and hexamethonium response, evaluation associated with sympathetic nervous system, as well as increased anti-inflammatory (IL-10/TNF-α ratio) and antioxidant (CAT, SOD and GPx) profiles in cardiac tissue when CET was associated with HCTZ, but not in SHR treated only with HCTZ.

Importantly, these autonomic, inflammatory and redox positive adaptations induced by CET plus HCTZ were associated with improvement in cardiac function, evaluated by RPP, an important measure of cardiac function, representing a direct indication of the energy demand of the heart and thus a good measure of the cardiac energy consumption. In fact, we observed reduced RPP in ovariectomized SHR treated with HCTZ or trained, and in the association group (OHT) when compared to ovariectomized group. However, only the groups submitted to CET (OT and OHT) showed lower RPP compared to hypertensive control females (C group). Similar results were observed in female SHR after aerobic exercise training [64]. Considering that our findings demonstrated that RPP was correlated with Var-SAP, LF-SAP, baroreflex sensitivity and IL-10/TNF-α ratio, we suggest that autonomic and inflammatory CET plus HCTZ induced-improvements contributed to reduce cardiac damage and to enhance cardiac function.

Additionally, we previously demonstrated that baroreflex sensitivity is inversely related to oxidative stress damage in ovariectomized SHR [28]. Furthermore, a negative relationship between baroreflex sensitivity and end-organ damage has been observed in male SHR [48]. In accordance with these findings, we found that the improvement in baroreflex sensitivity (bradycardic response) was associated with lower TNF-α and protein oxidation in cardiac tissue in ovariectomized SHR, reinforcing the role of autonomic control of circulation in the modulation of mechanisms related to end-organ damage in AH, such as inflammation and oxidative stress.

It is important to emphasized that a significant number of hypertensive females are treated, but their AP remains uncontrolled [2, 4]. In this sense, the non-regulation of key mechanisms involved in AH, as well as cardiometabolic impairments caused by chronic AH, can compromise continuum well-management of the disease. Furthermore, the doses of antihypertensive medications are related to the side effects [12], which are determinant factors for discontinuation of the treatment [65]. In this sense, low adherence affects both medication [4] and exercise [66] approaches in AH. However, in contrast to the side effects of HCTZ, exercise induces numerous beneficial physiological effects for the body. Moreover, regular exercise is beneficial in AH for a possible reduction in the amount and/or dose of medication or, in some cases, discontinuation of medication, and is a key approach for the well-controlled AP. Strategies proposed to improve exercise adherence in hypertensive individual has showed also promising results [67]. Thus, the overall message is that the intensification of the recommendation by health professionals, including clinicians, is crucial for the adoption and continuity of a physically active life, which can have long-term health and health care costs implications.

Finally, it is important to remind that the primary goal of AH treatment is to manage AP and AH-related consequences, such as end-organ damage. However, some pharmacological

therapy, despite lowering basal AP, were associated with activation of contra regulatory mechanism of AP control, such as sympathetic activity or RAAS. In this context, our findings support the exercise training as a coadjutant therapy in hypertensive postmenopausal women receiving HCTZ, implying a beneficial role of the combination of these approaches not only in AP control, but also in mechanisms associated with cardiac damage in AH. Future research should investigate the combination of exercise and antihypertensive drugs, with a focus on the effects on end-organ damage in different AH patients. Based on our data in the overall experimental and clinical evidence regarding the benefits of exercise training in hypertensive patients, we believe that this non-pharmacological approach, as well as an active lifestyle, should be encouraged and incorporated to a better management of classical and remaining risk in hypertensive patients on pharmacological treatment.

## Conclusions

We concluded that CET alone or in combination with HCTZ were more effective than HCTZ alone in improving baroreflex sensitivity and APV in a model of hypertension and postmenopause. Furthermore, the combination of HCTZ and CET resulted in additional positive adaptations in HRV and sympathetic tonus in the basal AP. Importantly, these autonomic benefits were linked to better inflammatory and redox balance in the heart, which is a target organ in AH. Thus, combining exercise with a medication approach could be a promising strategy for managing dysfunctions associated with classic and remaining cardiovascular risk in AH in postmenopause.

## Supporting information

**S1 Fig. Systolic arterial pressure assessed by tail plethysmography.**
(DOCX)

**S1 Table. Correlation analysis involving all studied groups.**
(DOCX)

**S1 File.**
(DOCX)

## Author Contributions

**Conceptualization:** Maycon Junior Ferreira, Maria Claudia Irigoyen, Kátia De Angelis.

**Data curation:** Kátia De Angelis.

**Formal analysis:** Maycon Junior Ferreira, Kátia De Angelis.

**Funding acquisition:** Maycon Junior Ferreira.

**Investigation:** Maycon Junior Ferreira, Michel Pablo dos Santos Ferreira Silva, Danielle da Silva Dias, Nathalia Bernardes.

**Methodology:** Maycon Junior Ferreira, Danielle da Silva Dias.

**Project administration:** Maycon Junior Ferreira.

**Supervision:** Kátia De Angelis.

**Validation:** Kátia De Angelis.

**Visualization:** Kátia De Angelis.

**Writing – original draft:** Maycon Junior Ferreira, Kátia De Angelis.

**Writing – review & editing:** Maycon Junior Ferreira, Maria Claudia Irigoyen, Kátia De Angelis.

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
