## [Decision Letter · Decision Letter 0]

24 Apr 2023

PONE-D-23-10192Concurrent exercise training induces additional benefits to hydrochlorothiazide: evidence for improve on autonomic control and oxidative stress in hypertensive ratsPLOS ONE

Dear Dr. De Angelis,

Thank you for submitting your manuscript to PLOS ONE. After careful consideration, we feel that it has merit but does not fully meet PLOS ONE’s publication criteria as it currently stands. Therefore, we invite you to submit a revised version of the manuscript that addresses the points raised during the review process.

We look forward to receiving your revised manuscript.

Kind regards,

Michael Bader

Academic Editor

PLOS ONE

Journal Requirements:

Reviewers' comments:

Reviewer's Responses to Questions

**Comments to the Author**

1. Is the manuscript technically sound, and do the data support the conclusions?

Reviewer #1: Partly

Reviewer #2: Partly

2. Has the statistical analysis been performed appropriately and rigorously? 

Reviewer #1: Yes

Reviewer #2: Yes

3. Have the authors made all data underlying the findings in their manuscript fully available?

Reviewer #1: Yes

Reviewer #2: Yes

4. Is the manuscript presented in an intelligible fashion and written in standard English?

Reviewer #1: Yes

Reviewer #2: No

5. Review Comments to the Author

Reviewer #1: In the current study the authors investigated the effect of exercise in combination with HCTZ in ovariectomized SHRs. The animal experiment mimics a relevant clinical situation in which postmenopausal females are at higher risk of arterial hypertension and subsequent hypertension-dependent end-organ damage.

The strength of the study is the analysis of the effect of exercise on autonomic cardiovascular control. It is important to show that this exercise effect is not only stable in co-treatment with HCTZ but also slightly more improved in the combination. As expected this goes along with reduced inflammation in the heart. The weakness of the current study is the lack of any endo-organ functional data such as heart function (LVEF etc) or renal function (like proteinuria etc.). It is also important to make clearer that the BP was not reduced in exercise performing rats but the age-dependent further increase was attenuated. Thus, exercise contributes to the subsequent further development of hypertension and related disease but is not sufficient to reduce blood pressure. In the clinical meta-analysis cited in the study (Ref. 15) all exercise performing patients had a reduced body weight and were over-weight. Therefore, unlike the well conducted animal study here, this reference is misleading and does also not share the common view of the literature in which a blood pressure reducing effect of SHRs is more critically discussed.

Main points:

N numbers of animals are slightly different per group. This suggests that individual animals were removed from further analysis. Please state in the study why this was done.

An interesting point not further mentioned or discussed by the reviewers is the lack of effect of exercise on IL6 expression. Is this because exercise performing SHRs had slightly bigger hearts?

Body weights and heart weights are reported. Unfortunately, the heart weight is properly wrong because a heart weight of 0.7 mg for an SHR is unlikely. The study would further benefit from a report of normalized heart weights to body weight (if possible also to tibia length which would be more accurate). This is really informative as there are no further reports about the heart function.

It remains unclear what the information about SOD in cardiac tissue should explain. Is this SOD2 or SOD1 or both and what about SOD in plasma.

A limitation of the study is the lack of inflammatory markers in plasma as hypertension affects blood vessels.

Reviewer #2: The paper entitled “Concurrent exercise training induces additional benefits to hydrochlorothiazide: evidence for improve on autonomic control and oxidative stress in hypertensive rats” is very interesting. So, I will give you the opportunity to revise some points before resubmitting to the journal.

Title: “Concomitant exercise training induces additional benefits to hydrochlorothiazide: evidence to improve autonomic control and oxidative stress in hypertensive rats”. There is an important point to be noted in the title that was not addressed, which would be postmenopausal hypertensive rats, not just hypertensive rats. I believe that as this study is based on looking for additional effects of HCTZ training in postmenopausal hypertensive animals, this should be considered when writing the title.

Abstract

Objective: “Although hydrochlorothiazide (HCTZ) is commonly prescribed to treat high blood pressure (BP), its role in restoring BP regulatory mechanisms is debatable”. “We focused to evaluate whether physical training could contribute to a better modulation of neurohumoral mechanisms in postmenopausal rats treated with HCTZ”. This session you should be more objective and less introductory to your topic.

Another point, the term high arterial pressure (AP) is not the ideal way to address arterial hypertension, because of the differences in having high blood pressure and being hypertensive. I suggest you review this concept.

Methods: “Spontaneously hypertensive rats (SHR) were distributed into sedentary (S) and ovariectomized groups: sedentary alone (OS), treated with HCTZ alone (OSH), trained alone (OT) or combined with HCTZ (OTH).” improve this sentence, as it is not well understood whether all animals are from the SHR model or the ovariectomized animals are normotensive, explain the number of animals and days of life. Given the name sedentary to the untrained SHR group, is not the correct word to say, because animals, even if not trained, can generate energy expenditure inside the cage, for example, so the most correct name, can only be the hypertensive group or hypertensive control/no intervention group. For the sedentary alone (OS) group, ovariectomized hypertensive patients seem to be more correct. What techniques were used to evaluate your objective? needs to be introduced to make more sense of the methods.

Keywords: "cardiovascular prevention", prevention or treatment, your animals already have hypertension and are being treated with hydrochlorothiazide, the use of the word prevention generates verbal disagreement with your methodology.

Introduction

It needs to be redone, the mechanisms studied in the article were not addressed in the introduction, what is the importance and influence of arterial hypertension on inflammation and oxidative stress? What does this imply in the regulation of autonomic mechanisms? How can the drug hydrochlorothiazide and exercise act on the regulation pathways of these autonomic and vasoconstrictor control mechanisms of arterial hypertension?

The use of acronyms in your article is not following the standards of publication with high blood pressure. 1- Arterial hypertension (HTN) this designation is incorrect the abbreviation HTN refers to hypertension in general, the more correct abbreviation term would be Arterial Hypertension (AH), emphasizing the type of hypertension studied. 2- arterial pressure (AP) or high arterial pressure (AP), is using the same abbreviation for two different things, in addition, if designating arterial hypertension as high arterial pressure is incorrect, because a person or animal that has arterial pressure high does not necessarily have the cardiovascular comorbidity of arterial hypertension.

In the sentence "The hydrochlorothiazide (HCTZ), a thiazide diuretic, has been one of the initial classes recommended in monotherapy or combined with other ntihypertensive drugs, being widely used in clinical practice", is this treatment recommended by whom? Does this information come from a guideline? What recommended dose? How many times a day can everyone take this medication? Improve the phrase and refer to it to provide scientific support.

In this sentence: "promoting significant clinical reduction around −12 and −6 mmHg for SAP and DAP, respectively (15)" the acronyms SAP and DAP were not described or mentioned earlier in the text, what do they mean?

In the sentence: "Despite this, exercise training induced chronic AP reduction is associated with adaptations in key AP control mechanisms such as autonomic, humoral, inflammation, and oxidative stress (17–21)". Necessary to be rewritten, for better emphasis and importance to the exercise. How does this happen? Which mechanism? Why study the baroreflex? Why study TNF-alpha, interleukins 6 (IL-6) and 10 (IL-10), how exercise modulates the pathway in hypertension? What is the importance of exercise in the redox state, does it decrease the increase of anti and pró oxidative enzymes? How is exercise related to vasoconstrictor or vasorelaxant mechanisms? How will all this together with hydrochlorothiazide treatment be beneficial? These points should be considered to base the objectives of your article, as they are mechanisms already studied and of great importance in the modulation of exercise and arterial hypertension.

Another point not addressed by the authors, what is the influence of hormone deprivation induced by ovariectomized hypertension?

Methodology

In this sentence: "Female spontaneously hypertensive rats (SHR) (150-200g, 90 days old) were obtained from Nove de Julho University (UNINOVE) (Sao Paulo, Brazil) and randomly allocated into (n=7−8 rats each group) sedentary (S), ovariectomized sedentary (OS), ovariectomized sedentary treated with hydrochlorothiazide (OSH), ovariectomized trained (OT) and ovariectomized trained and treated with hydrochlorothiazide (OTH)." Suggestion already mentioned about the denomination sedentary to groups not trained in physical exercise.

Why did the authors use young rats aged 90 days (~12-13 weeks old) instead of older animals, which are already considered models of hypertension and postmenopause? Fortepiane (2002) suggests that 18-month-old postmenopausal rats, but not ovariectomized rats, may be a suitable model for the study of postmenopausal hypertension. Furthermore, SHR rats reach a stable level of hypertension at around 17-19 weeks of age, so why use ~12-13-week rats? Where have animals of other ages been examined in previous studies? Ref: Lourdes A. Fortepiani et al, (2002) - Characterization of an Animal Model of Postmenopausal Hypertension in Spontaneously Hypertensive Rats - https://doi.org/10.1161/01.HYP.0000046924.94886. EF Hypertension. 2003;41:640–645.

"Pharmacological treatment was performed using hydrochlorothiazide (Sanofi Medley Farmacêutica, Campinas, SP, Brasil), an antihypertensive drug corresponding to thiazide diuretics class, at a dose of 30 mg/kg/day. According to our pilot study conducted previously, this dose during 1 wk showed sufficient to promote an AP reduction of approximately 10−12 mmHg in ovariectomized SHR", previously published scientific and methodological support is necessary, this dose or another has possibly already been used by other authors, who sought the purpose of treating arterial hypertension.

"The hydrochlorothiazide tablet was macerated, diluted in drinking (filtered) water, and then was made available for consumption" where has this methodology been applied before? What is the treatment efficiency of this injection method? How was the exact injection of 30 mg/kg/day controlled, since there were 2-4 animals per box and how each one consumed the exact dose stipulated by the authors? Is this method better than the commonly used gavage administration?

Did the authors consider the half-life of the drug hydrochlorothiazide used in this study? The half-life of administration of a single dose of hydrochlorothiazide varies between 6 and 9 hours, with a peak effect between the first 4 and 6 hours of injection, with a maximum total duration of 12 hours, when considering interval doses, which are composed of two daily doses half-life ranges from 8 to 15 hours, with peak effect not determined, duration of effect 16 to 24 hours. But the animals ingested varying amounts of water with the compound during the day, how is the efficiency of this treatment ensured? It was described by the authors "The daily consumption was monitored and then considered to adjust the amount of water for the groups undergo to the drug treatment." Does this method ensure that all animals consume 30 mg/kg/day? How this was done, I need better explanations of this method.

Extensive revision would be necessary in view of the abbreviations used in the manuscript, as many of them are mentioned without using the full name, thus making reading unfeasible.

In the topic " Inflammatory mediators", How was the tissue prepared for the analysis of inflammatory gauges, as the authors only describe that the heart was collected and that a commercially available ELISA kit (R&D Systems Inc.) was used to assess levels of TNF-alpha and interleukins 6 (IL-6) and 10 (IL-10) in cardiac tissue using microplate method. Was the tissue homogenized? Was it used in its entirety? How were these samples processed?

These sentences "The recorded data were analyzed on a beat-to-beat basis to quantify changes in SAP, diastolic (DAP), mean AP (MAP), and HR (19,22).", due to not using correct abbreviations without prior description, it is not known the purpose of the technique or what data it provides.

There is a lack of a topic that addresses anthropometry used in this study, listing which organs were collected and processed, since in view of the results, the authors exchanged data on animal weights, orb weight, adipose tissue weight, and none of this is listed in the methodology of this study.

Results and discussion

In all the topics present in the discussion, I advise the authors to start the paragraphs, explaining the importance of the analysis and the reason for it, so that they have a more fluid line of reasoning. As an example in the topic "Baroreflex sensitivity " the authors start the topic as follows: "Both trained groups (OT and OTH) presented an increased bradycardic response to phenylephrine compared with the OS group (OT: −1.3 ± 0.4 and OTH: −1.6 ± 0.3 vs. OS: −0.6± 0.3 bpm/mmHg) (p < 0.001)", the die is just thrown in the topi without a beginning of content, it would be more fluid if it started " before the analysis of the sensitivity of the baroflex, we could notice that ..."

In front of the topic "Anthropometry", in this topic the sutores present the results in a descriptive way, saying for example "However, weight gain during the study was higher in rats treated with HCTZ alone (OSH vs. S group)", but they do not present the values nor how much was the difference between the groups, I suggest identifying in percentage for example. The data in this topic are being sent as supplementary material, however I think it necessary to add the table as solid manuscript results, as they are discussed in the next topic. In the same table of the supplementary material for data on muscles such as soleus and Plantaris, which, when discussed in the topic, are described only as skeletal muscle, I suggest that they be listed, since each of these muscles has a different function and has different metabolisms, they are activated differently in each of the types of exercises performed in this present study.

This topic needs to be rewritten so that it can resemble the others, together with its data it should appear during the topic and not as supplementary material.

In the topic "Tests of maximum exercise", the following sentence: "For the maximal running test, there were time (p < 0.001), group (p < 0.021) and interaction effects (p < 0.001). " Needs modifications, as cannot be understood, it was not possible to understand the intention and function of the sentence.

In this same topic, I suggest that the figure in front of these data are not explained through supplementary material, I believe that these data are of great value for the consolidation and understanding of the results exposed here, being part of the main data of your study.

In the topic "Tail plethysmography", I once again reinforce the need to verify all the acronyms present in this article, so that each one of them is fully described.

In the discussion the authors say "However, we found that the non-exercised HCTZ-treated group gained more body weight and had more white adipose tissue weight (see S1 Table). In contrast, body weight gain and white adipose tissue were similar in control and trained groups, suggesting that the edication approach can induce anthropometric changes even in the absence of increased feed intake", how is it possible to explain weight gain in the non-exercised HCTZ-treated group, considering that food consumption did not show differences? organism, by which route would the amount of adipose tissue increase?

The paragraphs "HCTZ has been linked to metabolic disorders, such as insulin resistance and fat storage (12). In our study, we did not ..." and "Performance in the treadmill exercise test improved in both trained groups, as previously reported (18), but sig ...", do not have a discussion, not the explanation of the mechanism, the authors only report again the data found. In this discussion topic it is necessary that the data be compared with others found in the literature.

Following the discussion, the authors say: "he initial dose chosen in monotheraphy to achieve adequate control of AP using HCTZ in hypertensive individuals is 25mg daily. Inadequate AP control after medication treatment, on the other hand, is common, and strategies such as increasing the dose and/or frequency, concurrent use of diuretics, or combination with other classes of antihypertensive are used." Who recommends these doses? Why is it necessary to change the dose? how does it affect increasing the daily dose or fractioning it for example? These responses are very important to support the choice of dose used in this article, what do the guidelines for the treatment of arterial hypertension say about this drug, what is the best dose, what frequency? are essential answers to be addressed in the present study and in the present discussion.

In this paragraph " The baroreceptor reflex regulates moment-to-moment the AP and plays a key role for maintaining cardiovascular homeostasis. Exercise training has previously been shown to improve baroreflex sensitivity in female ovariectomized rats (17,18,22). In the present study, we showed that ovariectomy reduced the tachycardic responses of baroreflex sensitivity. However, CET alone or combined with HCTZ improved both bradycardic and tachycardic responses of baroreflex sensitivity. The bradycardic response is more vagal dependent, while the tachycardic responses is more sympathetic-dependent. The increase and decrease of the vagal and sympathetic activity, respectively, has been cumulatively supported as a positive adaptation to exercise training. We hypothesize that improvements in vagal modulation, as measured by the increase in HF band and indirectly assessed by HRV, contributed to improvements in bradycardic responses. Although our study found no improvement in vagal modulation in the OT group as measured by HRV, it is possible that CET alone increased vagal tonus to the heart (32), as evidenced by the expressive resting bradycardia seen in the OT group. In the same way, it is possible that reduction in cardiac, vascular and overall sympathetic contribution, as verified by the reduction of the LF-PI (OTH group), LF-SAP (OT and OTH groups) and the lower reduction of the MAP after hexamethonium injection (OTH group), respectively, support the enhanced tachycardic responses". on the reflex baroreceptor? How does deprivation of hormones impact on the reflex baroreceptor? How does exercise along with pharmacological treatment impact on the reflex baroreceptor?

In the following sentence "However, the influence of the HCTZ on sympathetic activity are inconsistent. HCTZ has been related to increased response to supine and upright MSNA (34)." What would MSNA be? Once again here I say, it would take an extensive review of the abbreviations. Also, does this sentence make no sense? Does it need to be rewritten?

The following sentence "HTN is associated with end-organ damage. Although AP level management is the focus of treatment, its variability is another factor that requires attention due to its contribution to end-organ damage. High APV has been linked to decreased and increased cardiac (39) and renal function (39) and injury (40), as well as all-cause mortality (41)." This is displaced from the discussion, after introductory data, and does not enter the scope of the article when we stop to think about results related to this work.

The authors need to correct the references, as some are non-standard or follow different formats, for example in the sentence "We previously show that anti-inflammatory cytokines after ovariectomy, and that aerobic and resistance exercise training improved renal and that aerobic and resistance exercise training improved renal 18 and cardiac inflammation in ovariectomized SHR (18,20)."

" Considering the anti-inflammatory effect of exercise inseveral diseases that has been widely described in the literature (44), and based on ourdata, it is plausible to speculate that exercise training contributed more to this adaptation in the OTH group than HCTZ", Which cytokines? what mechanisms? how does aerobic or anaerobic exercise modulate these anti-inflammatory pathways? How is this important for arterial hypertension?

The authors wrote that "We found high pro-oxidant (NADPH oxidase) reduced antioxidant CAT enzyme activity and increased protein damage (carbonyls) in cardiac tissue of the OS group", however the phrase is meaningless, making it difficult to understand, I believe it is a problem in the translation from English, it would be necessary to correct the English in relation to this manuscript.

"Evidence evaluating the effect of HCTZ on oxidative stress showed inconsistent findings." Where was this published? Which authors?

"Importantly, CET combined with HCTZ reduced NADPH oxidase and hydrogen peroxide, enhanced CAT, SOD and GPx activities and decreased protein oxidation in cardiac tissue." What mechanism of activation of these enzymes in the face of the redox state induced by physical exercise?

"Inflammatory mediators are known to active pro-oxidants and influence redox balance. In this regard, the current study found that an increase in cardiac IL-10/TNF-α was associated with lower protein oxidation levels. ." Here the authors resumed saying the same thing as said above, about inflammation and the redox state, it would be better to unite the paragraphs and generate more links between these data.

"Antihypertensive medications have greater hypotensive effects at higher doses. Their side effects, however, are dose proportional. About this, tolerability may be compromised, and the individual may choose to discontinue treatment in the long run." Who described this previously? Where is it published? What do the guidelines say about?

Finally, I leave the following points as needs of this article, in addition to those already highlighted above:

1- correction of acronyms

2- English correction

3- Improvement of the discussion, as it seems like an instruction, as there is no discussion with data already published in the literature

6. PLOS authors have the option to publish the peer review history of their article (what does this mean?). If published, this will include your full peer review and any attached files.

Reviewer #1: No

Reviewer #2: No

---

## [Author Response · Author response to Decision Letter 0]

16 Jun 2023

The authors would like to thank the reviewers for their valuable comments. We carefully considered all of the suggestions and accomplished with all of the requests within the aim and possibilities of the present study. The responses to the questions are in the file 'Response to Reviewers', attached along with the revised article.

---

## [Decision Letter · Decision Letter 1]

20 Jul 2023

PONE-D-23-10192R1Concurrent exercise training induces additional benefits to hydrochlorothiazide: evidence for improve on autonomic control and oxidative stress in a model of hypertension and postmenopausePLOS ONE

Dear Dr. De Angelis,

Thank you for submitting your manuscript to PLOS ONE. After careful consideration, we feel that it has merit but does not fully meet PLOS ONE’s publication criteria as it currently stands. The new title is grammatically incorrect since "improve" is a verb. Suggestion: Concurrent exercise training induces additional benefits to hydrochlorothiazide: evidence for an improvement of autonomic control and oxidative stress in a model of

hypertension and postmenopause

We look forward to receiving your revised manuscript.

Kind regards,

Michael Bader

Academic Editor

PLOS ONE

Journal Requirements:

Please review your reference list to ensure that it is complete and correct. If you have cited papers that have been retracted, please include the rationale for doing so in the manuscript text, or remove these references and replace them with relevant current references. Any changes to the reference list should be mentioned in the rebuttal letter that accompanies your revised manuscript. If you need to cite a retracted article, indicate the article’s retracted status in the References list and also include a citation and full reference for the retraction notice

Reviewers' comments:

Reviewer's Responses to Questions

**Comments to the Author**

1. If the authors have adequately addressed your comments raised in a previous round of review and you feel that this manuscript is now acceptable for publication, you may indicate that here to bypass the “Comments to the Author” section, enter your conflict of interest statement in the “Confidential to Editor” section, and submit your "Accept" recommendation.

Reviewer #1: All comments have been addressed

2. Is the manuscript technically sound, and do the data support the conclusions?

Reviewer #1: Yes

3. Has the statistical analysis been performed appropriately and rigorously? 

Reviewer #1: Yes

4. Have the authors made all data underlying the findings in their manuscript fully available?

Reviewer #1: Yes

5. Is the manuscript presented in an intelligible fashion and written in standard English?

Reviewer #1: Yes

6. Review Comments to the Author

Reviewer #1: I have no further comments. However, the authors may take more attention on limitations of their study and remind that IL6 is a myokine rather than a pro-inflammatory cytokine in the context of execise.

7. PLOS authors have the option to publish the peer review history of their article (what does this mean?). If published, this will include your full peer review and any attached files.

Reviewer #1: No

---

## [Author Response · Author response to Decision Letter 1]

20 Jul 2023

The authors would like to thank the Academic Editor and Reviewer for their important considerations. We considered the suggestions for the present study. Responses are provided below, and changes made in the manuscript are highlighted.

Academic Editor:

The new title is grammatically incorrect since "improve" is a verb. Suggestion: Concurrent exercise training induces additional benefits to hydrochlorothiazide: evidence for an improvement of autonomic control and oxidative stress in a model of hypertension and postmenopause.

We appreciate the important observation of the Academic Editor. We rewrote the new title following the suggestion.

Reviewer #1: 

I have no further comments. However, the authors may take more attention on limitations of their study and remind that IL6 is a myokine rather than a pro-inflammatory cytokine in the context of exercise.

We appreciate the valuable comments from the reviewer. We acknowledge the potential limitations as well as the strengths of our study. At this time, we will consider the comments to further enhance our results and interpretations.

---

## [Editor Report · Decision Letter 2]

24 Jul 2023

Concurrent exercise training induces additional benefits to hydrochlorothiazide: evidence for an improvement of autonomic control and oxidative stress in a model of hypertension and postmenopause

PONE-D-23-10192R2

Dear Dr. De Angelis,

We’re pleased to inform you that your manuscript has been judged scientifically suitable for publication and will be formally accepted for publication once it meets all outstanding technical requirements.

Kind regards,

Michael Bader

Academic Editor

PLOS ONE
---

## [Editor Report · Acceptance letter]

28 Jul 2023

PONE-D-23-10192R2 

Concurrent exercise training induces additional benefits to hydrochlorothiazide: evidence for an improvement of autonomic control and oxidative stress in a model of hypertension and postmenopause 

Dear Dr. De Angelis:

I'm pleased to inform you that your manuscript has been deemed suitable for publication in PLOS ONE. Congratulations! Your manuscript is now with our production department. 

Kind regards, 

on behalf of

Prof. Michael Bader 

Academic Editor

PLOS ONE